# Occupancy estimation of wild species in a palm oil plantation using unstructured data

**Arco J. van Strien[1], Erik Meijaard[2]\*, Nardiyono[3], Syafiie Suief[2], Syahmi Zaini[2]**

**1** Arco van Strien Data Analysis, Leiden, The Netherlands, **2** Borneo Futures, Bandar Seri Begawan, Brunei Darussalam, **3** Austindo Nusantara Jaya, Jakarta, Indonesia

\* emeijaard@gmail.com

## Abstract

In 2020–2024, plantation workers at a large oil palm plantation in West Kalimantan recorded sightings of wild species of several species groups. They did not use a standardized field method and produced no comprehensive reports but recorded only one or a few species per visit. Such unstructured data are generally viewed as challenging for statistical analysis. Here, we used occupancy models to estimate what proportion of spatial units (both non-natural and forest) a given species occupied annually in the plantation. We tested models with different covariates for species for which most data were available: 13 birds, 3 reptiles and 7 mammals, among which the iconic Orangutan (*Pongo pygmaeus*). For each species, the model which best fitted the data was selected. Two shortcomings in our data complicated the analyses. First, occupancy models require detection and non-detection records, but because no fixed species lists were used, there is no unique way to generate non-detections. We generated non-detection records in several ways and ran the models again to evaluate the consequences for occupancy estimates. Second, imbalances in sampling may occur because of a lack of sampling design to select study sites. Most concerning are sites surveyed only once in 2020–2024. We ran the models without those sites to examine whether results were different. Although the shortcomings mentioned turned out not to distort our results, the occupancy estimates were imprecise for many study species because of low detection rates, and extra efforts are needed to improve that.

## Introduction

The expansion of oil palm (*Elaeis guineensis* Jacq.) plantations in tropical areas affects wildlife negatively [1–3]. Over time, the palm oil industry has been increasingly willing to mitigate the negative effects by taking protection and management measures. Examples are setting aside areas with High Conservation Values (HCV-areas) which are protected against illegal logging, burning and land-clearing [4] and the protection of "forest islands" within plantations [5].

---

**Data availability statement:** All relevant data are available from Zenodo (https://doi.org/10.5281/zenodo.17595516).

**Funding:** AJvS and EM received funding from Austindo Nusantara Jaya to develop the citizen science pilot study and statistical methods for the company's estates. Data collection was done by volunteers from the Austindo Nusantara Jaya workforce. The funders had no role in data analysis or preparation of the manuscript.

**Competing interests:** The authors have declared that no competing interests exist.

Effective biodiversity protection and management measures require detailed knowledge about local wildlife, but collecting standardized data on many species by trained field workers is usually not feasible in tropical areas [6]. Furthermore, protection and management measures are more likely to succeed if there is considerable support among plantation workers and other stakeholders. We believe that obtaining more data and increasing support can go hand in hand by encouraging plantation workers and others to collect sighting records of wild species.

For these two reasons, in 2019, the Indonesian oil palm company Austindo Nusantara Jaya (ANJ) launched a special program, called PENDAKI — the Indonesian acronym for Care for Biodiversity — to stimulate the recording of sightings of wild animals and plants by plantation workers in their concessions [6]. One of these areas is the KAL ("Kayung Agro Lestari") oil palm estate in West Kalimantan. This estate contains hundreds of production blocks planted with oil palms and several small forest islands (total area 9,583 ha; Fig 1). The estate also includes several large HCV-areas (together 3,845 ha) with remnants of the original tropical moist forest. Since 2019, many observers have participated in the program to collect wildlife data, both in the planted areas and in the HCV-areas. To enable everyone to participate, there are hardly any rules that observers must adhere to. For each sighting, location (identity of production block or forest site), date and species must be noted, as well as the number of individuals observed. Apart from this, there is no prescribed, standardized field method and most of the records are casual sightings.

To make these data usable for the management of plantations, the collected data need to be converted into relevant information, such as local differences in occurrence of species or in species diversity, which may then be related to environmental factors or management practices. Unfortunately, retrieving reliable information from casual sightings is challenging. Data as collected here are referred to in the scientific literature as unstructured or opportunistic data [8,9]. Such data are collected by volunteer observers in many countries around the world and are also known as citizen science data [10].

Because the field method is unstandardized, it is difficult to determine whether the difference in occurrence between two locations is real or due to one location being more intensively surveyed than the other. For unstructured data, it is therefore mandatory to adjust for variation in field effort. This is not an easy task, especially if there is no information at all available about the field effort applied, as in the current study. It is difficult to derive inferences on abundance of species from unstructured data [though see 11], but it is often feasible to estimate occupancy, *i.e.*, the proportion of spatial units occupied by the species. Analyses of unstructured data are particularly successful when occupancy models are used [8,12–14]. But not all applications have been fruitful [9], and much depends on the dataset available.

Our ultimate goal is to apply citizen science data with occupancy models on a routine basis to support wildlife management in large plantations in tropical regions [6]. This is an ambitious goal, and, as far as we know, we are the first to scientifically explore this. Here we report on the first step towards that goal by examining the quality of the occupancy estimates. The aim of this paper is to develop custom occupancy

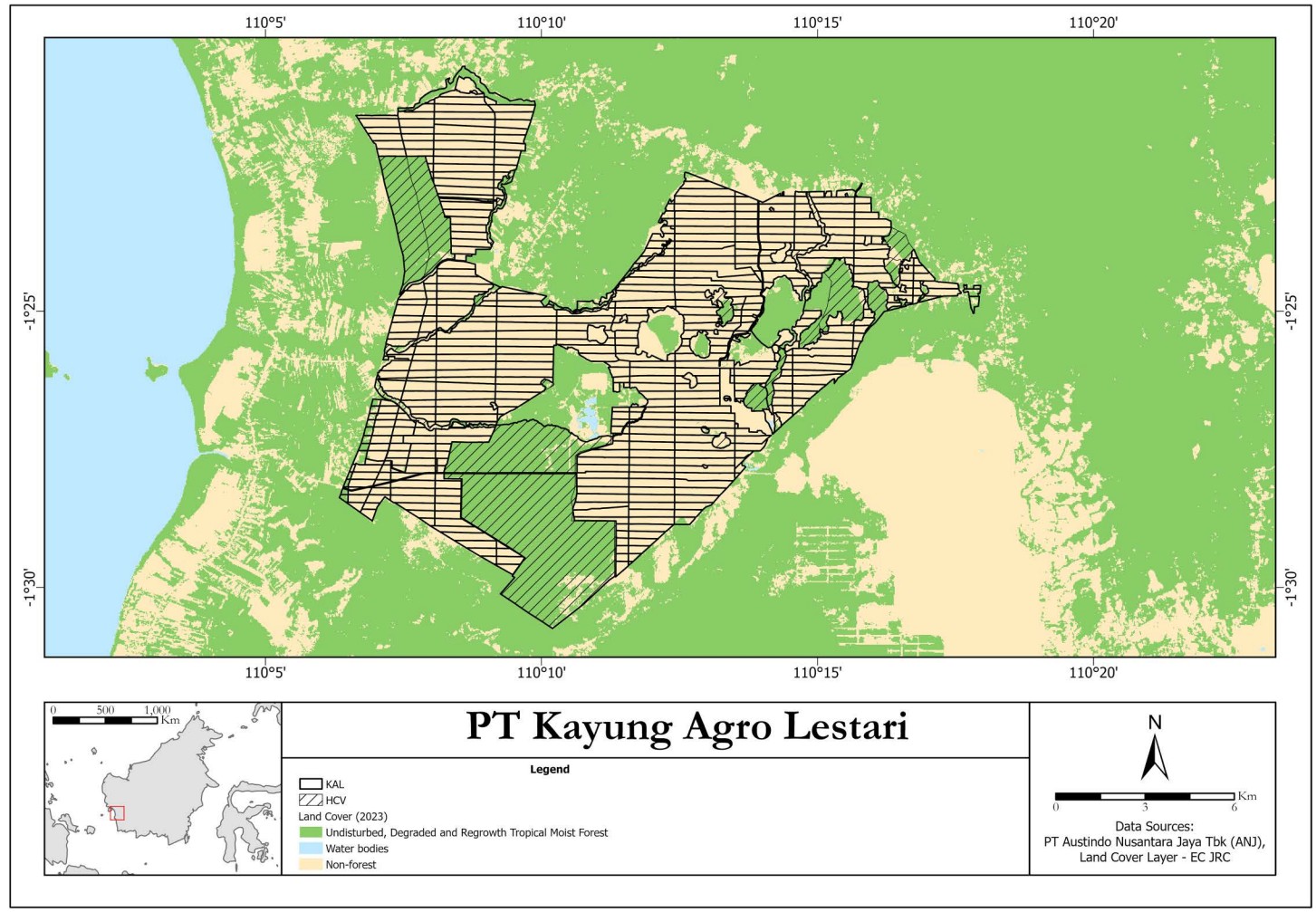

**Fig 1. Map of the KAL estate in West Kalimantan, Indonesia using the Tropical Moist Forest (TMF) baseline dataset produced by the European Commission's Joint Research Centre (EC JRC) [7].** TMF data were generated under studies funded by the Directorate-General for Climate Action through the Roadless-For pilot project and the ForMonPol project. Data are provided free of charge under the Copernicus Regulation. Proper attribution follows JRC requirements (forobs.jrc.ec.europa.eu/TMF/data). TMF dataset described in Vancutsem et al. [7].

models for species living in an oil palm plantation landscape and to test whether several complications in the data might distort the results.

This study exemplifies the analysis of opportunistic data, an emerging field of research. Globally, the number of opportunistic data records has increased significantly through data portals such as iNaturalist (https://www.inaturalist.org), Online databases like GBIF (https://www.gbif.org) and eBird (https://ebird.org) already contain many millions of records and also cover a significant portion of historical data from museum collections and protected nature areas. It is therefore expected that opportunistic citizen science data will play an increasingly important role in biodiversity research, for example in assessments for the Red List of Threatened Species [15]. Applications can only be successful if the statistical weaknesses of the data are considered, and fortunately, in recent years significant progress has been made to address these issues [e.g., 11,13,16]. Most applications using unstructured data concern temperate regions of the Northern Hemisphere; applications in tropical regions are still scarce.

## Materials and methods

### Inclusivity in global research

Additional information regarding the ethical, cultural, and scientific considerations specific to inclusivity in global research is included in the Supporting Information (S1 Checklist).

### Area and data

Defining the production blocks and forest sites in the KAL plantation area as our sampling sites, we distinguish 463 sites in the KAL-area, 21 of which are completely covered with forest. The latter are not only the large HCV areas mentioned earlier, but also smaller forest islands within the estate which are not in use for palm oil production (Fig 1). The other sites (from now on referred to as non-natural sites) are dominated by oil palms and infrastructure, but may also contain ponds, waterways and small patches of forest regrowth. The surface area of the non-natural sites is 30 ha on average but varies between sites (Fig 1). Most forest sites are considerably larger, especially the HCV-sites, which unfortunately could not be split into smaller subsites because sightings in the HCV-sites were not precisely localized.

More than 400 observers provided data. Observers were allowed to record any species group they wanted. But because most records cover five species groups only — namely birds, mammals, reptiles, higher plants (*i.e.*, only *Magnoliopsida*) and insects (mainly butterflies and dragonflies) — we focused on these groups, abandoning data on other groups. This is our standard dataset, in which records are available from 387 sites for the 2020–2024 period. For about 200 sites, annual records are available (Table 1). On average, sites were visited 6–8 times per year (see Nvisits per site in Table 1), often by different observers who typically reported only one or a few species (rarely > 10) per visit (see List Length in Table 1). Some sites were never surveyed; these are mainly situated in recent plantings at the western and eastern border of the estate. Records of the same species on the same date and site collected by different observers were deduplicated because these may refer to the same sighting as observers are frequently in the field together.

### Occupancy models

Occupancy models aim to estimate occupancy of a species while considering the detection probability [14]. This means that a different detection probability of a species due to differences in, e.g.*,* field effort, observer experience or species abundance, does not translate into a different occupancy estimate of the species [12]. This can be particular useful for unstructured data because these models may still be able to properly estimate occupancy even when no standardized field method has been used and no information about field effort is available [8,12,13]. To be able to separate occupancy from detection, occupancy models consist of two hierarchically coupled Generalized Linear Models (GLMs), one to describe the true state of sites (presence or absence) and one to describe the detection in sites, given that the species does occur [14].

Table 1. Annual field effort in the KAL standard dataset.

| Year | Nsites | Nrecs | Nvisits | List Length |
|---|---|---|---|---|
| 2020 | 191 | 3047 | 8.3 | 1.9 |
| 2021 | 232 | 4055 | 7.1 | 2.4 |
| 2022 | 139 | 4462 | 7.4 | 4.3 |
| 2023 | 185 | 3348 | 7.5 | 2.4 |
| 2024 | 269 | 2567 | 6.0 | 1.6 |

Nsites: number of sites with observations, Nrecs: total number of presence records, Nvisits: mean number of visits per site, List Length: mean number of all observed species per visit.

The occupancy submodel estimates the annual proportion of occupied sites in the statistical population represented by the sampled sites. We estimated annual occupancy using a multi-season model for single species [17]. Our most comprehensive submodel is specified as follows:

$$z_{it} \sim \text{Bernoulli}\left(\psi_{it}\right) \tag{1}$$

$$\text{logit}\left(\psi_{it}\right) = \text{intercept} + \text{year} + \text{habitat of site} + \text{distance to large forest} + \text{site size} \tag{2}$$

where $z_{it}$ is the estimated occupancy (0 or 1) and $\psi_{it}$ represents the occupancy probability of the species in site $i$ and year $t$. This submodel includes covariates that describe the differences in occupancy between sites: year, habitat of the site, distance of the site to large forest and site size. Habitat of the site can be forest or non-forest; forest is taken when the site is a forest site or a non-natural site with a forest patch. Distance to large forest represents the distance of the center of a site to the nearest large forest areas; these are either forests around the estate or the HCV-areas within the estate. Site size is included to adjust for variation in surface area of the sites. Our main interest is in the annual occupancy estimates which we assessed for each species by summing the estimated $z_i$ values over sites and year [finite sample occupancy; 14].

The detection submodel estimates detection per visit and the most comprehensive model is specified as:

$$y_{ijt} \sim \text{Bernoulli}\left(p_{ijt} \times z_{it}\right) \tag{3}$$

$$\text{logit(detection)} = \text{intercept} + \text{list length} + \text{habitat of observation} + \text{observer} \tag{4}$$

in which the observations $y_{ijt}$ indicate that the species was detected ($y_{ijt} = 1$) or not ($y_{ijt} = 0$) at location $i$ during visit $j$ in year $t$. The observations are considered as the product of occupancy $z_{it}$ and the probability $p_{ijt}$ of detecting the species in location $i$ during visit $j$ in year $t$. [14]. This submodel includes covariates that describe the differences in detection between sites and visits: list length, observer and habitat of observation. There are no year effects in this model because data were too sparse to estimate reliable detection estimates which vary between the years. Length of species lists per visit was used as covariate because we expected lower detections in shorter lists. We did not apply list length categories [as in 8], but used list length as a linear detection predictor. Habitat of the observation can be forest or non-forest; forest is taken when the observation was in a forested part of a site; a species may be more difficult to spot there. Observer identity was modelled as a random effect to take into account the variation in detection between observers [18]. Estimating detection $p$ is only possible if repeated visits are available for at least some sites [14]. The replicated visits need to be done in a period of closure, meaning that the state of all locations (occupied or not occupied) has not changed during the period between the visits. In contrast to studies using European data [e.g., 8], we here consider an entire calendar year as the closure period, thereby assuming that the study species reside in the area all year round. This does not necessarily mean that a species is present in a site during the entire year, but rather that a species may use the site year-round [14]. Our preliminary analyses showed no effect of Julian date on detection, so we did not include Julian date as a covariate in the model.

Occupancy models require detection and non-detection records while our dataset only contains presence records (detections) of five species groups. We created a non-detection record for all species of these five groups not being observed during a field visit [8]. We ran the models in a Bayesian framework using the R-package *spOccupancy* [19] in R v. 4.3.3 [20]. All quantitative covariates (distance to large forest, site size and list length) were standardized to have mean 0 and standard deviation 1 [19]. *spOccupancy* allows accounting for spatial autocorrelation of occupancy sites, which may be useful for our study area where sites are adjacent to each other. Unfortunately, the large inaccuracy in the locations of

observations within the large HCV-sites made it impossible to apply a spatially explicit model. With the covariate distance-to-large-forest, however, we included a variable in the model describing spatial gradients in the study area.

More than 300 different species of the five species groups were reported in 2020–2024, but for most of them too few presence records had been collected to allow statistical analysis. Models were only run for species with at least five presence records each year and more than 150 records in total. These were 23 species: 13 birds, 7 mammals and 3 reptiles. To find the model which best fitted the data, we ran 18 different models per species. In these models, one or several covariates were omitted from the model described by [2] and [4] (see S2 Table for an overview of the 18 models). The model with the lowest value of the Widely Applicable Information Criterion (WAIC) was considered the best model [19]. For these top models (S3 Table) we performed posterior predictive checks to compute Bayesian *p*-values as a measure of model fit. More specific, following the recommendation of Doser & Kéry [21], we used both the Freeman-Tukey statistic and the Chi-Square statistic, and grouped the data across sites as well as across replicates. Ideally, each Bayesian *p*-value is around 0.5.

We adopted the default priors of *spOccupancy* and ran the models with three parallel Markov chains of 60,000 iterations each, discarding the first 50,000 as burn-in and a thinning rate of 10. A parameter was considered significant when the 95% Bayesian credible interval (CRI) of the posterior sample of a parameter did not include 0. The standard deviation of the sample from the posterior distribution of parameter was interpreted as standard error of that parameter.

## Alternative datasets

We created several alternative datasets to study the effect of two complications in the data and applied the same 18 models as for the standard dataset. Firstly, observers did not make complete checklists of species groups, nor did they use a fixed list of species to consider. Consequently, there is no unique way to generate non-detection records. If an observer only noticed, say, plant species, and ignored any sighting of the other four species groups, non-detections generated for other species groups are not justified. To examine if this affects occupancy estimates, we created three alternative datasets differing in the way non-detections were created: (1) we filtered data from the three most reported groups (birds, mammals and reptiles) and discarded records from higher plants and insects. This implies that we generated non-detections for our 23 study species only from sightings of birds, mammals and reptiles and no longer from plants and insects, (2) in addition to using only data from these three species groups, we included only observers with records of the same species group as the study species. So, if the study species were a mammal, only sightings of observers reporting mammals were used to generate non-detections and the same goes for birds and reptiles; (3) in addition to using data from three species groups, we included only observers who had recorded all three species groups, thereby further reducing the number of observers (Table 2). The three extra datasets are called (1) Nondet3grp, (2) Nondet3grp+obs1grp and

**Table 2. Properties of datasets per species obtained under different procedures (see text for explanation).**

| Dataset | Nsites | Nvisits | List Length | Nobservers | N detect. | Nnon-detect. |
|---|---|---|---|---|---|---|
| Standard data | 387 | 7.2 | 2.4 | 372 | 534 | 6757 |
| Nondet3grp | 386 | 7.0 | 2.2 | 367 | 534 | 6499 |
| Nondet3grp+obs1grp | 382 | 7.1 | 2.3 | 255 | 534 | 6336 |
| Nondet3grp+obs3grp | 374 | 7.2 | 2.3 | 132 | 512 | 6135 |
| Remove_sites1year | 279 | 7.8 | 2.4 | 365 | 523 | 6570 |
| Remove_sites2024 | 340 | 7.4 | 2.4 | 370 | 530 | 6681 |

Nsites: number of sites with observations, Nvisits: mean number of visits per site, List Length: mean number of all observed species per visit, Nobservers: number of observers, Ndetect. and Nnon-detect.: mean number of detection and non-detection records of the 23 study species.

(3) Nondet3grp+obs3grp and occupancy results were compared with those of the standard dataset. All filtering actions reduced the number of non-detections, and to a lesser extent also the number of detections (Table 2).

Secondly, unstructured data such as our dataset not only lack a standardized field method but also have no sampling design to properly select study sites. In our case, forest sites and non-natural sites surrounding forest sites have been oversampled, so the measurements might not be representatives for the entire estate [22]. On the other hand, data came from almost the entire area, which does not suggest obvious spatial bias. There is, however, an imbalance in sampling which we are concerned about: 108 sites (28% of all 387 sites) have only been surveyed in one year, most frequent in 2024. If the differences in occupancy between 2020 and 2024 would depend on the sites that were surveyed only once, then we have no convincing evidence of a change. To check this, we created two more datasets: one in which all sites surveyed in one year only were deleted (Remove_sites1year) and one in which sites visited only in 2024 were deleted (Remove_sites2024), and compared annual occupancies. Obviously, these two datasets contain fewer sites than the standard dataset (Table 2).

## Results

### Standard dataset

Almost all models applied to the standard dataset converged according to the Gelman-Rubin Rhat statistic (Rhat < 1.1). Only nine out of 414 models (18 different models * 23 species) did not converge in all parameters, most frequently not in case of *Spilornis cheela,* a bird of prey species. The top model according to the WAIC-value varied between species (Table 3). For several species, the most comprehensive model (model 1 in S2 Table) fitted best, for example in case of *Hylobates albibarbis* (gibbon) and *P. pygmaeus.* For other species, simpler models performed better, *e.g.*, for *Liopeltis tricolor*, a snake species. The top model for the latter species contained only effects of site size, species list length and observer and no ecological variables at all. The Bayesian *p*-values produced by three out of the four tests for the top models were mostly between 0.2–0.75, suggesting a reasonable fit. But for half of the species *p*-values < 0.2 were obtained according to the Freeman-Tukey test when data were grouped across sites (Table 3).

Detection probability varied between species, with *Varanus salvator* (a large monitor lizard) having the highest rate of 0.26 (Table 3). This means that *V. salvator* has been reported during one-quarter of the visits to occupied sites. Here, a visit stands for a single visit of an average observer collecting a species list of mean length outside forest habitat. For most species detection is low, frequently even below 0.1 (Table 3). Detection increased with longer species lists (Table 3; see example in Fig 2A). In line with our expectation, many species appeared to be less visible in forest habitat, such as *Centropus sinensis*, a cuckoo species (Table 3), although this may also be because these species spend less time in forest habitat. In contrast, some others were better detectable in forest habitat or spend more time in forest habitat, *e.g.*, *Macaca nemestrina* (a monkey species). Furthermore, observer effects were relevant for all species, as we already suspected. Models without observer effects consistently finished low in the model selection.

Mean probability of occupancy of *V. salvator* is 0.97 (Table 4), suggesting that this species makes use of nearly the entire study area. *P. pygmaeus*, for example, has a much lower mean occupancy of 0.26. Several species benefitted from the availability of forest habitat in the site including *M. nemestrina*, *Macaca fascicularis* and *P. pygmaeus* (Table 4). Furthermore, the distance to the nearest large forest affected the occupancy probability for about half of the species. Not surprisingly, *H. albibarbis* had a significantly higher occupancy in sites with forest habitat and in the vicinity of large forest areas (Table 4; Fig 2B), and the same applies to *P. pygmaeus*. Some other species, however, achieved their highest occupancy rate further away from the large forest, *e.g.*, *Spilopelia chinensis*, a dove species. For almost all species, occupancy was higher in larger sites (Table 4). Fig 3 shows examples of annual occupancy, for *P. pygmaeus* and *Amaurornis phoenicurus* (a waterbird species).

**Table 3. Detection parameter estimates under the top model per species.**

| Species | Model ID | Bayesian p-value | p (detection) | List Length | Habitat of observation | Observer |
|---|---|---|---|---|---|---|
| **Birds** | | | | | | |
| Acridotheres javanicus | 8 | 0.32 | 0.08 ± 0.01 | 0.69* | −1.25* | 1.00* |
| Alcedo meninting | 1 | 0.07 | 0.02 ± 0.01 | 0.73* | −0.4* | 3.33* |
| Amaurornis phoenicurus | 8 | 0.11 | 0.09 ± 0.01 | 0.67* | −1.59* | 0.77* |
| Anthracoceros malayanus | 10 | 0.16 | 0.03 ± 0.01 | 0.48* | | 1.23* |
| Centropus sinensis | 8 | 0.34 | 0.17 ± 0.01 | 0.63* | −0.63* | 0.37* |
| Corvus enca | 13 | 0.27 | 0.02 ± 0.01 | 0.46* | | 2.14* |
| Egretta garzetta | 1 | 0.18 | 0.07 ± 0.01 | 0.49* | −0.98* | 1.25* |
| Elanus caeruleus | 12 | 0.10 | 0.06 ± 0.01 | 0.6* | | 1.24* |
| Gracula religiosa | 1 | 0.08 | 0.06 ± 0.01 | 0.46* | −0.59* | 0.51* |
| Halcyon smyrnensis | 14 | 0.51 | 0.06 ± 0.01 | 0.69* | −1.02* | 1.31* |
| Hirundo tahitica | 11 | 0.38 | 0.02 ± 0.01 | 0.41* | −1.43* | 2.01* |
| Spilopelia chinensis | 11 | 0.19 | 0.14 ± 0.01 | 0.59* | −1.15* | 0.38* |
| Spilornis cheela | 13 | 0.28 | 0.01 ± 0.01 | 0.67* | | 2.69* |
| **Reptiles** | | | | | | |
| Liopeltis tricolor | 15 | 0.31 | 0.01 ± 0.01 | 0.7* | | 2.14* |
| Malayopython reticulatus | 2 | 0.37 | 0.05 ± 0.02 | 0.49* | −1.37* | 3.10* |
| Varanus salvator | 14 | 0.17 | 0.26 ± 0.02 | 0.86* | −0.31* | 0.67* |
| **Mammals (rodents)** | | | | | | |
| Callosciurus notatus | 7 | 0.61 | 0.04 ± 0.01 | 0.65* | | 0.78* |
| Callosciurus prevostii | 3 | 0.23 | 0.02 ± 0.01 | 0.54* | 0.39* | 1.35* |
| Nannosciurus melanotis | 1 | 0.03 | 0.05 ± 0.01 | 0.57* | −0.39* | 1.72* |
| **Mammals (monkeys)** | | | | | | |
| Hylobates albibarbis | 1 | 0.06 | 0.04 ± 0.01 | 0.47* | 1.22* | 2.20* |
| Macaca fascicularis | 7 | 0.01 | 0.16 ± 0.02 | 0.59* | | 1.56* |
| Macaca nemestrina | 1 | 0.02 | 0.07 ± 0.01 | 0.57* | 0.37* | 1.31* |
| Pongo pygmaeus | 1 | 0.08 | 0.06 ± 0.02 | 0.55* | 0.99* | 2.93* |

See S2 Table for the meaning of model IDs. Bayesian *p*-values are a measure of model fit according to the Freeman-Tukey statistic with data grouped by sites [21]. *p* is mean detection probability in 2020–2024 (± se), *i.e.,* the backtransformed intercept as specified in formula [4]. Coefficients of list length, habitat of observation and observer are on the logit scale. No entry indicates that the covariate is not included in the model. Asterisks refer to significant differences from zero, derived from 95% credible intervals.

## Alternative datasets

The models converged somewhat less often in the alternative datasets than in the standard dataset. To enable comparisons between datasets, we did not delete any of the species from the results. Often but not always in the alternative dataset, the same model was selected as top model as in the standard set (S3 Table). Mean detection in 2020–2024 derived from the three datasets with alternative procedures to generate non-detections was significantly higher than when based on the standard dataset (Fig 4A; paired t-tests; P < 0.05; n = 23). This is not unexpected because the three alternative datasets all have fewer non-detection records than the standard data (Table 2). Mean occupancy estimates in 2020–2024, however, did not differ between those derived from the standard dataset and those based on the three alternative datasets (Fig 4B; paired t-tests; P > 0.05; n = 23). Thus, the occupancy estimates do not appear to depend on the exact way non-detections were generated.

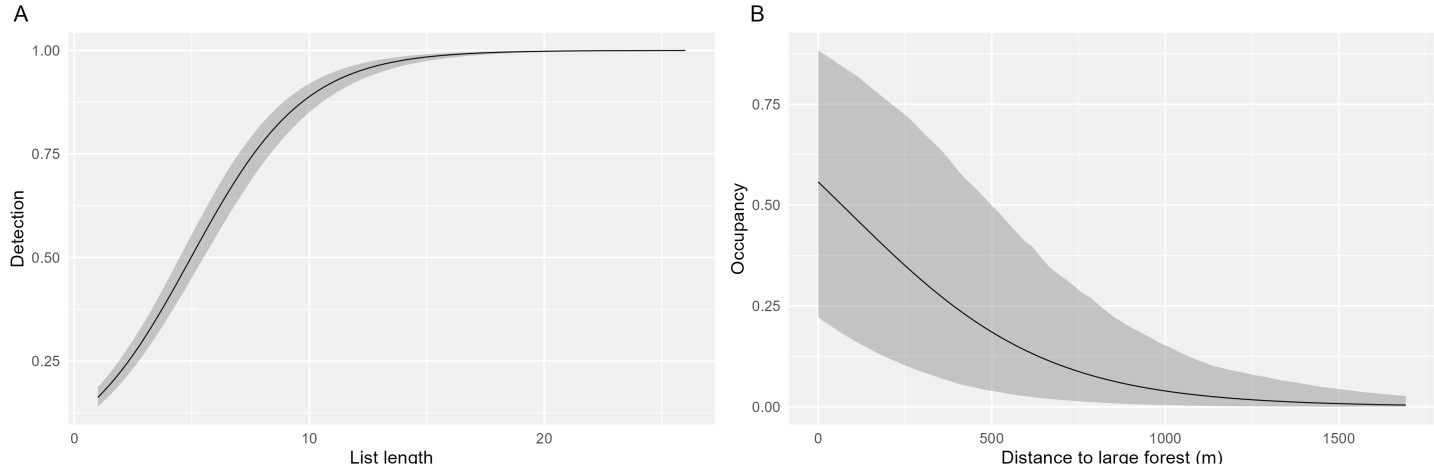

**Fig 2. Relationship between (A) expected detection (± credible intervals) of *Varanus salvator* and the number of species observed during a visit and (B) between expected occupancy of *Hylobates albibarbis* and distance to large forest.**

The two datasets for studying the removal of poorly surveyed sites produced similar detection and occupancy estimates as found for the standard dataset ([Fig 4A] and [4B]; paired t-test; P > 0.05; n = 23). Also, differences between 2020 and 2024 derived from these datasets were strongly correlated with those based on the standard dataset (Pearson's r > 0.99; P < 0.01; n = 23).

## Discussion

Many researchers have pointed out the downsides of the collection and analysis of unstructured data: bias when field effort is not taken into account, spatial bias due to lack of sampling design, low detection when field efforts are limited and uncertainty about non-detections when it is not sure which species are being considered [e.g., [16],[23]]. In some citizen science projects, these shortcomings are reduced by discouraging the collection of casual records and instead focusing on complete checklists, *i.e.*, lists of all species observed and identified by a participant at a particular time and place [e.g., eBird, see [24]]. This provides more clarity about non-detections, and higher detection probabilities. If more information about the observation process is provided, for example time spent or length of route during the visit, there are more opportunities in statistical analyses to adjust for field effort [[25],[26]]. With a smartphone app such information may be generated automatically [[16]].

The collection of data in the KAL area is slowly moving in this direction. A subgroup of observers has recently been selected to use a fixed list of species which should be recorded when observed [[6]]. The hope is that this will lead to longer species lists. This group will work with a smartphone app that can record the spatial coordinates of each observation, which makes it possible to derive field efforts to some extent. The availability of spatial coordinates makes it possible to localize observations on a small scale afterwards. This will solve one of the obvious shortcomings of the current KAL data, namely the overly large forest sites.

Yet, it is not the intention to limit data collection to fixed lists only. On the contrary, the idea remains to encourage as many workers as possible to participate in data collection, to promote awareness about and involvement in the protection of biodiversity. There may still be considerable potential, because there are many observers who contributed only a few records so far. That leaves us little choice but to deal with all shortcomings of the data. We have shown that the occupancy estimates were unaffected by the way non-detection records were generated. The fact that many records were collected by a relatively small group of dedicated observers can partly explain this, as they collected records from all five

**Table 4. Occupancy parameter estimates under the top model per species (see Table 3 for model ID's).**

| Species | ψ | SE | Habitat of site | Distance to forest | Site size |
|---|---|---|---|---|---|
| **Birds** | | | | | |
| Acridotheres javanicus | 0.65 | 0.09 | | | 1.83* |
| Alcedo meninting | 0.83 | 0.07 | −1.21 | 0.45* | 1.34* |
| Amaurornis phoenicurus | 0.84 | 0.06 | | | 1.21 |
| Anthracoceros malayanus | 0.53 | 0.12 | 1.11 | −0.61* | |
| Centropus sinensis | 0.84 | 0.05 | | | 1.71* |
| Corvus enca | 0.30 | 0.11 | 0.99 | | 2.04* |
| Egretta garzetta | 0.69 | 0.08 | −0.40 | 0.44* | 2.28* |
| Elanus caeruleus | 0.55 | 0.08 | | −0.64* | 1.42* |
| Gracula religiosa | 0.24 | 0.07 | 1.47* | −0.38 | 2.14* |
| Halcyon smyrnensis | 0.57 | 0.09 | | | |
| Hirundo tahitica | 0.59 | 0.09 | | 0.3 | |
| Spilopelia chinensis | 0.90 | 0.05 | | 1.15* | |
| Spilornis cheela | 0.52 | 0.18 | 2.00* | | |
| **Reptiles** | | | | | |
| Liopeltis tricolor | 0.41 | 0.12 | | | 2.23* |
| Malayopython reticulatus | 0.38 | 0.10 | | −0.44* | 2.02* |
| Varanus salvator | 0.97 | 0.02 | | | 1.83* |
| **Mammals (rodents)** | | | | | |
| Callosciurus notatus | 0.48 | 0.09 | 0.64 | −0.66* | 1.55* |
| Callosciurus prevostii | 0.63 | 0.14 | 1.22 | | 1.63* |
| Nannosciurus melanotis | 0.33 | 0.07 | 1.47* | −0.31 | 2.05* |
| **Mammals (monkeys)** | | | | | |
| Hylobates albibarbis | 0.33 | 0.08 | 1.33* | −1.22* | 1.53* |
| Macaca fascicularis | 0.39 | 0.05 | 1.61* | −0.58* | 1.69* |
| Macaca nemestrina | 0.47 | 0.09 | 1.10* | −0.89* | 1.56* |
| Pongo pygmaeus | 0.26 | 0.07 | 1.22* | −0.59* | 2.08* |

ψ is the mean finite sample annual occupancy, averaged over 2020–2024. SE is the standard error of annual occupancy, also averaged over 2020–2024. Coefficients of habitat of site, distance to forest and site size are on the logit scale. No entry indicates that the covariate is not included in the model. Asterisks refer to significant differences from zero, derived from 95% credible intervals.

groups. We also showed that visiting sites only once did not influence the results. It is encouraging that the models yielded meaningful relationships between occupancy and distance to large forest and forest habitat preference. These relationships confirm existing knowledge [e.g., 27,28], thereby enhancing confidence in the usefulness of the data being collected in the KAL estate. With longer time series data, it may be interesting to study whether these relationships are changing over time. An increase in occurrence further away from large forest areas may indicate greater use of non-natural sites and/or higher permeability of the plantation. Such changes have been demonstrated for orangutans in Malaysian oil palm who increasingly make use of oil palm areas around forest fragments [29].

But methodological problems remain, the biggest one being the short species lists in our data (Table 1). This leads to the generation of many non-detections (one for each species not observed) which induces low detection rates per visit. Although there are quite a few replicated visits per site (Table 1), these do not fully compensate for the low detection, and the result is that occupancy estimates for many species are far from precise [14]. Using simulation data, Kery & Royle [30; p. 588] showed that the precision of the occupancy estimates becomes low

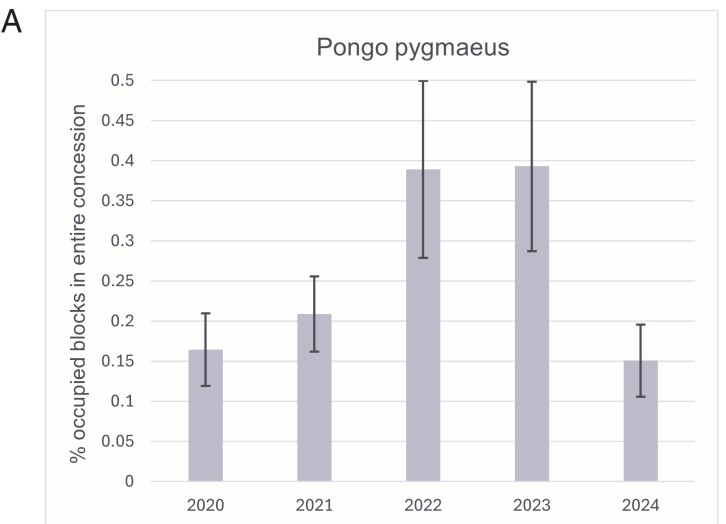
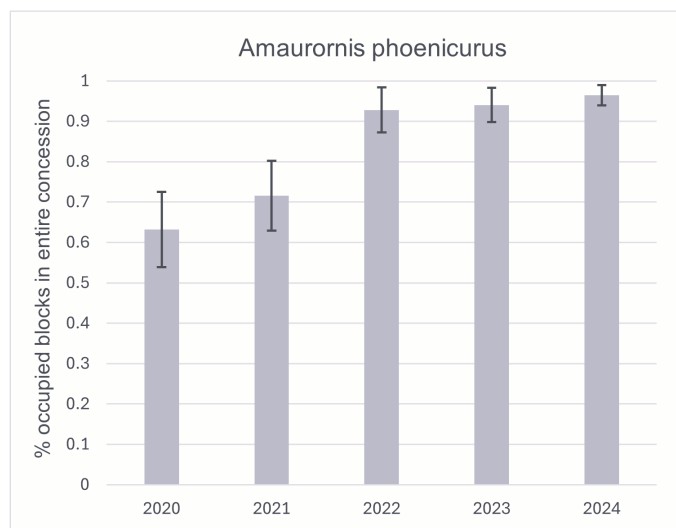

**Fig 3. Annual occupancy (± se) of (A)** *Pongo pygmaeus* **and (B)** *Amaurornis phoenicurus.*

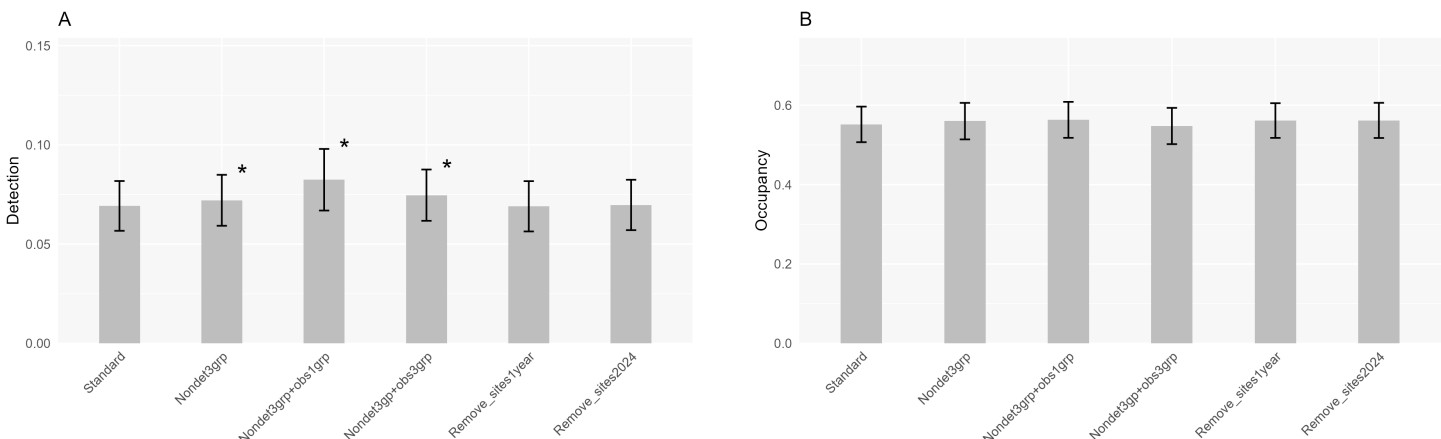

**Fig 4. Comparison of key statistics between different models. (A)** Mean detection (± se; backtransformed intercept; see text) and **(B)** mean occupancy (± SE) in 2020–2024 of 23 species. Each bar represents a different dataset. Asterisk indicates significant difference with standard dataset (paired t-test; P < 0.05; n = 23).

when *p* is less than about 0.1–0.2. Most of our species have detection rates < 0.2 and indeed, annual occupancy estimates are quite imprecise, with standard errors of annual occupancy > 0.05 or even > 0.1 (Table 4). To improve this, more data are needed, preferably in the form of longer species lists. Another problem is that model fit was not yet sufficient for all species, which contributes to the uncertainty of inferences. The lack-of-fit test revealed that spatial variation in occupancy and detection is not yet adequately modeled for a number of species [Table 3; 21]. Model fit might be enhanced by including more adequate covariates as well as spatial autocorrelation in the model. With more annual data it also becomes more feasible to include year-dependent detection estimation. So far, we assumed detection probability was equal in all five years. But it is better to consider potential changes in detection, for example because of increasing average skill of observers [12].

In conclusion, although we have made progress in model development and solved several data complications, the quality of the occupancy estimates is not yet sufficient to be useful for the nature management of the plantation. More data is needed to obtain more precise occupancy estimates for the 23 study species and for additional species. More data will also make spatially more refined applications feasible. We are confident that we will achieve this in the long run.

## Supporting information

**S1 Checklist. Inclusivity in global research.**
(DOCX)

**S2 Table. Overview of models.** Model 1 is the most comprehensive model described in the main text. The others are models in which one or more covariates have been omitted. No entry indicates that the covariate is not included in the model.
(DOCX)

**S3 Table. Top model per species per dataset according to WAIC.** See S2 Table for the meaning of model IDs.
(DOCX)

## Acknowledgments

Thanks are due to all the employees of Austindo Nusantara Jaya (ANJ) in the KAL estate who collected the wildlife data and facilitated the implementation of the PENDAKI program. The KAL-database is managed by Priya Swayanuar and Claudia Retina Munthe. Fig 1. was created by Safwanah Ni'Matullah. We thank Willy van Strien for critically reading the manuscript, and two peer reviewers for their constructive feedback.

## Author contributions

**Conceptualization:** Arco J. van Strien, Erik Meijaard.

**Data curation:** Arco J. van Strien, Erik Meijaard, Syafiie Suief.

**Formal analysis:** Arco J. van Strien, Syahmi Zaini.

**Investigation:** Arco J. van Strien, Erik Meijaard, Nardiyono, Syahmi Zaini.

**Methodology:** Arco J. van Strien, Erik Meijaard.

**Resources:** Erik Meijaard.

**Supervision:** Erik Meijaard, Nardiyono.

**Validation:** Erik Meijaard, Syafiie Suief, Syahmi Zaini.

**Visualization:** Syafiie Suief, Syahmi Zaini.

**Writing – original draft:** Arco J. van Strien, Erik Meijaard, Syafiie Suief, Syahmi Zaini.

**Writing – review & editing:** Arco J. van Strien, Erik Meijaard.

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
