## [Decision Letter · Decision Letter 0]

10 Oct 2025

Dear Dr. Meijaard,

Thank you for submitting your manuscript to PLOS ONE. After careful consideration, we feel that it has merit but does not fully meet PLOS ONE’s publication criteria as it currently stands. Therefore, we invite you to submit a revised version of the manuscript that addresses the points raised during the review process.

We look forward to receiving your revised manuscript.

Kind regards,

Edvard Mizsei

Academic Editor

PLOS ONE

Journal Requirements:

“AJvS and EM received funding from Austindo Nusantara Jaya to develop the citizen science pilot study and statistical methods for the company's estates.”

5. We note that you have indicated that there are restrictions to data sharing for this study. PLOS only allows data to be available upon request if there are legal or ethical restrictions on sharing data publicly. For more information on unacceptable data access restrictions, please see http://journals.plos.org/plosone/s/data-availability#loc-unacceptable-data-access-restrictions.

6. We note that Figure 1 in your submission contain map/satellite images which may be copyrighted. All PLOS content is published under the Creative Commons Attribution License (CC BY 4.0), which means that the manuscript, images, and Supporting Information files will be freely available online, and any third party is permitted to access, download, copy, distribute, and use these materials in any way, even commercially, with proper attribution. For these reasons, we cannot publish previously copyrighted maps or satellite images created using proprietary data, such as Google software (Google Maps, Street View, and Earth). For more information, see our copyright guidelines: http://journals.plos.org/plosone/s/licenses-and-copyright.

7. We notice that your supplementary tables are included in the manuscript file. Please remove them and upload them with the file type 'Supporting Information'. Please ensure that each Supporting Information file has a legend listed in the manuscript after the references list.

Reviewers' comments:

Reviewer's Responses to Questions

**Comments to the Author**

1. Is the manuscript technically sound, and do the data support the conclusions?

Reviewer #1: No

Reviewer #2: Yes

2. Has the statistical analysis been performed appropriately and rigorously?

Reviewer #1: No

Reviewer #2: Yes

3. Have the authors made all data underlying the findings in their manuscript fully available?

Reviewer #1: Yes

Reviewer #2: No

4. Is the manuscript presented in an intelligible fashion and written in standard English?

Reviewer #1: Yes

Reviewer #2: Yes

Reviewer #1: The study tackles an important conservation question: how far citizen science observations can inform species occupancy in human-modified tropical landscapes. The effort to compile and analyse such a large dataset is commendable, and the manuscript is generally well written.

However, I do not consider the current analysis to provide reliable or actionable ecological conclusions.

Detection probability and data quality:

Many species show very low detection probabilities (<0.1), resulting in wide confidence intervals and highly uncertain occupancy estimates. Such limitations cannot be addressed by simple manuscript revisions; they would require different data collection strategies (standardised sampling).

Model fit and spatial limitations:

The lack of precise site level coordinates (especially within HCV areas) further constrained the use of spatially explicit models. These shortcomings compromise the validity of the inferred habitat relationships.

Restricted species set:

Although over 300 species were recorded, occupancy analysis was workable for only 23 of them. This narrow focus reduces the generality of the conclusions for biodiversity management.

Interpretation of results:

The manuscript sometimes overstates the practical management implications of the findings. The data currently highlight the challenges of using opportunistic observations rather than delivering strong conservation guidance.

Presentation:

The writing is clear.

I therefore recommend rejection of the manuscript in its present form.

Reviewer #2: The objective of the MS is to explore novel methods of using N-mixture models to estimate occupancy by wildlife in particular habitats, using unstructured data obtained from citizen science surveys/occasional observations. With the increasing role of citizen science data collection in biodiversity sciences, this study is particularly important to explore how these unstructured data can be exploited to better understand the reaction of wildlife to alterations of natural habitats into landscapes serving human needs. In particular, the MS intends to overcome the problem of missing non-detection records. This manuscript is highly relevant in light of the quickly accumulating unstructured biodiversity data globally: via mobile applications (iNaturalist and others), data liberation from past and current publications via machine reading, from museum collections, data accumulating in data aggregators (i.e. GBIF etc.). The analyses of these data will require new and refined methods such as the one presented in this MS.

I consider this MS suitable for publication in PLOS ONE with a minor revision.

Language: As a non-native English speaker, I, in general, find the language clear and easy to understand. However, in some parts, the text would benefit from further polishing by a native speaker.

Remark on the Introduction and Discussion

The authors’ scope of the usefulness of their study, in my opinion, is a bit narrow. Their approach and methods are applicable in a wider scope (see my introductory overview). I recommend reviewing the literature shortly to include other fields of application in addition to what they indicated. The published paper would greatly benefit from opening up the scope of applicability (at least at the level of mentioning with a few references): historical miscellaneous data extracted from past literature, national park data repositories, data from museum specimens or biodiversity data aggregators etc.

Tables and figures:

Table 4: It would be helpful for the reader to group the observed taxa into higher taxonomic groups: Reptilie/Reptiles, Aves/Birds, Mammalia/Mammals and indicate the category in a subheading above each group. Maybe even at the level of order it would be reasonable to group them, e.g. not to mix primates with rodents. Also in S2.

In the legends and headings of tables 1,2. the names of the variables should be self-explanatory and given in a standard way: Nsites = No. of sites, etc. Also symbols or words should be standardised across the table legends/headings: either /(slash) or per.

Fig 3A-B, scientific names should be Italicised.

Minor issues:

Line78: References outside the sentence!

Line83: Instead of intensively studied it would be better to say intensively surveyed or sampled.

Lines118-119: “Some sites which were never surveyed;” “which” is not needed.

Lines 135-138: I have no deep understanding of the model, but this sentence seems to me in any case vague. It should be clearer for the benefit of the lay readers.

Line266: Varanus as a generic should be abbreviated: V. as it is the second mention in the MS. See also Pongo in Line280. See also guidelines of PLOS ONE.

Line286, Parenthesis after dove species should be deleted.

Line293: Consider rewording into: Often but not always in the alternative dataset, the same model was selected as top model as in the standard set.

Lines298, 303 and 305: data should be dataset.

Line309: Consider sentence revision: Many researchers pointed out the downsides of the collection and analysis of unstructured data:

Remark: The authors indicated that they have not have yet the full consent from the data owners to make the raw data available. Whatever the final decision will be the MS is still worth publishing.

**Do you want your identity to be public for this peer review?** For information about this choice, including consent withdrawal, please see our Privacy Policy

Reviewer #1: No

Reviewer #2: No

---

## [Author Response · Author response to Decision Letter 1]

11 Dec 2025

Response to reviewers

Reviewer #1: The study tackles an important conservation question: how far citizen science observations can inform species occupancy in human-modified tropical landscapes. The effort to compile and analyse such a large dataset is commendable, and the manuscript is generally well written.

However, I do not consider the current analysis to provide reliable or actionable ecological conclusions.

Response: That is correct. We do not yet provide ecological conclusions relevant for management purposes. In the MS, we stated that delivering conservation guidance is our ultimate, long-term goal. The current study is only the first step towards that goal and examines the quality of the occupancy estimates.

We have adapted our formulation to clarify the goal: “Here we report on the first step towards that goal by examining the quality of the occupancy estimates”

Detection probability and data quality:

Many species show very low detection probabilities (<0.1), resulting in wide confidence intervals and highly uncertain occupancy estimates. Such limitations cannot be addressed by simple manuscript revisions; they would require different data collection strategies (standardised sampling).

Response: Our occupancy estimates are indeed less precise than desired. We agree with the reviewer that this cannot be treated by adjusting the manuscript. However, we disagree with the reviewer that only standardized sampling offers a solution here. Although standardized sampling does provide higher data quality, for various reasons, it is not a feasible solution. We explain this in more detail in our simultaneously submitted manuscript (Maharani et al.; referred to in the MS).

We have clarified this shortly in the manuscript with the following addition (underlined) and refer further to Maharini who described this in more detail:

Effective biodiversity protection and management measures require detailed knowledge about local wildlife, but collecting standardized data on many species by trained field workers is usually not feasible in tropical areas (Maharani et al. 2025).

Model fit and spatial limitations:

The lack of precise site level coordinates (especially within HCV areas) further constrained the use of spatially explicit models. These shortcomings compromise the validity of the inferred habitat relationships.

Response: We disagree. As we stated in the MS, we were unable to included spatial autocorrelation in the model due to the lack of georeferenced observations within the HCV-areas. We only included the area of grid cells and the nearest distance of observations outside HCV-areas to the border of HCV-areas as covariates in the model. These covariates were not distorted by the lack of georeferenced observations within the HCV-areas, so we do not see why the inferred habitat relationships should be invalid.

Restricted species set:

Although over 300 species were recorded, occupancy analysis was workable for only 23 of them. This narrow focus reduces the generality of the conclusions for biodiversity management.

Response: It's true that inferences were possible for 23 of the more than 300 observed species. However, these already include some of the most iconic species found in the area (Orangutan and Gibbon). Considering just these species in the management would already be a gain. The goal, of course, is to make inferences possible for more species as the dataset increases. Furthermore, the exact same shortcoming applies to virtually any data collection, including standardized sampling.

Interpretation of results:

The manuscript sometimes overstates the practical management implications of the findings. The data currently highlight the challenges of using opportunistic observations rather than delivering strong conservation guidance.

Response: We agree that we should not overstate the management implications in this stage. In the MS, we already stated that delivering conservation guidance is our long-term goal. The current study is only the first step towards that goal and examines the quality of the occupancy estimates. We have adapted our formulation to make this clearer:

In conclusion, although we have made progress in model development and solved several data complications, the quality of the occupancy estimates is not yet sufficient to be useful for wildlife management of the plantation. More data is needed to obtain more precise occupancy estimates for the 23 study species and for additional species. More data will also make spatially more refined applications feasible. We are confident that we will achieve this in the long run.

Presentation:

The writing is clear.

I therefore recommend rejection of the manuscript in its present form.

Reviewer #2: The objective of the MS is to explore novel methods of using N-mixture models to estimate occupancy by wildlife in particular habitats, using unstructured data obtained from citizen science surveys/occasional observations. With the increasing role of citizen science data collection in biodiversity sciences, this study is particularly important to explore how these unstructured data can be exploited to better understand the reaction of wildlife to alterations of natural habitats into landscapes serving human needs. In particular, the MS intends to overcome the problem of missing non-detection records. This manuscript is highly relevant in light of the quickly accumulating unstructured biodiversity data globally: via mobile applications (iNaturalist and others), data liberation from past and current publications via machine reading, from museum collections, data accumulating in data aggregators (i.e. GBIF etc.). The analyses of these data will require new and refined methods such as the one presented in this MS.

Response: Thanks for the encouraging comment

I consider this MS suitable for publication in PLOS ONE with a minor revision.

Language: As a non-native English speaker, I, in general, find the language clear and easy to understand. However, in some parts, the text would benefit from further polishing by a native speaker.

Response: Thanks. We have checked the English carefully again.

Remark on the Introduction and Discussion

The authors’ scope of the usefulness of their study, in my opinion, is a bit narrow. Their approach and methods are applicable in a wider scope (see my introductory overview). I recommend reviewing the literature shortly to include other fields of application in addition to what they indicated. The published paper would greatly benefit from opening up the scope of applicability (at least at the level of mentioning with a few references): historical miscellaneous data extracted from past literature, national park data repositories, data from museum specimens or biodiversity data aggregators etc.

Response: We thought the potential benefits of unstructured data have already been sufficiently described elsewhere, but it's worth emphasizing them here again. We've therefore added a paragraph at the end of the introduction:

This study exemplifies the analysis of opportunistic data, an emerging field of research. Globally, the number of opportunistic data records has increased significantly through data portals such as iNaturalist (https://www.inaturalist.org). Large databases like GBIF (https://www.gbif.org) and eBird (https://ebird.org) now contain many millions of records and also cover a significant portion of historical data from museum collections and protected nature areas. It is therefore expected that opportunistic citizen science data will play an increasingly important role in biodiversity research, for example in assessments of Red Lists of threatened species (Maes et al. 2015). Applications can only be successful if the statistical weaknesses of the data are considered, and fortunately, in recent years significant progress has been made to address these issues (e.g., Isaac et al. 2014; Altwegg & Nichols 2019; Johnston et al. 2025). Most applications using unstructured data concern temperate regions of the Northern Hemisphere; applications in tropical regions are still scarce.

Tables and figures:

Table 4: It would be helpful for the reader to group the observed taxa into higher taxonomic groups: Reptilia/Reptiles, Aves/Birds, Mammalia/Mammals and indicate the category in a subheading above each group. Maybe even at the level of order it would be reasonable to group them, e.g. not to mix primates with rodents. Also in S2.

Response: we adapted table 3, table 4 and S2

In the legends and headings of tables 1,2. the names of the variables should be self-explanatory and given in a standard way: Nsites = No. of sites, etc. Also symbols or words should be standardised across the table legends/headings: either /(slash) or per.

Response: done

Fig 3A-B, scientific names should be Italicised.

Response: the scientific names were omitted from the figure and are now only available (in italic) in the figure legends.

Minor issues:

Line78: References outside the sentence!

Response: adapted

Line 83: Instead of intensively studied it would be better to say intensively surveyed or sampled.

Response: adapted

Lines 118-119: „Some sites which were never surveyed;” „which” is not needed.

Response: adapted

Lines 135-138: I have no deep understanding of the model, but this sentence seems to me in any case vague. It should be clearer for the benefit of the lay readers.

Response: We deleted this sentence entirely and added elsewhere:

“This submodel includes covariates that describe the differences in occupancy between sites:…”

“This submodel includes covariates that describe the differences in detection between sites and visits: …”

Line 266: Varanus as a generic should be abbreviated: V. as it is the second mention in the MS. See also Pongo in Line280. See also guidelines of PLOS ONE.

Response: done

Line 286, Parenthesis after dove species should be deleted.

Response: adapted

Line 293: Consider rewording into: Often but not always in the alternative dataset, the same model was selected as top model as in the standard set.

Response: adapted

Lines 298, 303 and 305: data should be dataset.

Response: adapted

Line 309: Consider sentence revision: Many researchers pointed out the downsides of the collection and analysis of unstructured data:

Response: adapted

---

## [Decision Letter · Decision Letter 1]

13 Jan 2026

Occupancy estimation of wild species in a palm oil plantation using unstructured data

PONE-D-25-36925R1

Dear Dr. Meijaard,

We’re pleased to inform you that your manuscript has been judged scientifically suitable for publication and will be formally accepted for publication once it meets all outstanding technical requirements.

Kind regards,

Edvard Mizsei

Academic Editor

PLOS One

Additional Editor Comments (optional):

Reviewers' comments:

Reviewer's Responses to Questions

**Comments to the Author**

Reviewer #2: All comments have been addressed

2. Is the manuscript technically sound, and do the data support the conclusions?

Reviewer #2: Yes

3. Has the statistical analysis been performed appropriately and rigorously?

Reviewer #2: Yes

4. Have the authors made all data underlying the findings in their manuscript fully available?

Reviewer #2: Yes

5. Is the manuscript presented in an intelligible fashion and written in standard English?

Reviewer #2: Yes

Reviewer #2: I found the topic and the promising method valuable. Even if the data has shortcomings what the authors admit, the methods explored here are promising, especially in light of the increasingly available unorganised data (citizen science, data liberation from literature, museum specimens). I hope that the authors will have the opportunity to further develop their methods.

**Do you want your identity to be public for this peer review?** For information about this choice, including consent withdrawal, please see our Privacy Policy

Reviewer #2: No

---

## [Editor Report · Acceptance letter]

PONE-D-25-36925R1

PLOS One

Dear Dr. Meijaard,

I'm pleased to inform you that your manuscript has been deemed suitable for publication in PLOS One. Congratulations! Your manuscript is now being handed over to our production team.

Kind regards,

on behalf of

Dr. Edvard Mizsei

Academic Editor

PLOS One